# Association between dementia risk and ulcerative colitis, with and without colectomy: a Swedish population-based register study

Miguel Garcia-Argibay [1,2] Ayako Hiyoshi,[1,3] Scott Montgomery [1,4,5]

¹Clinical Epidemiology and Biostatistics, School of Medical Sciences, Faculty of Medicine and Health, Örebro University, Orebro, Sweden
²Department of Medical Epidemiology and Biostatistics, Karolinska Institute, Stockholm, Sweden
³Department of Public Health Sciences, Stockholm University, Stockholm, Sweden
⁴Department of Epidemiology and Public Health, University College London, London, UK
⁵Clinical Epidemiology Division, Department of Medicine, Karolinska Institutet, Solna, Sweden

**Correspondence to**
Dr Miguel Garcia-Argibay;
miguel.garcia-argibay@oru.se

## ABSTRACT

**Objective** This study aims to investigate the association of ulcerative colitis (UC) with all-cause dementia and assess differences in those with and without a total colectomy.

**Design, setting and participants** This Swedish prospective register-based study comprised 4.8 million individuals aged at least 59 years between 1964 and 2018 with the linkage of several Swedish national registers.

**Primary and secondary outcome measures** Individuals with dementia were defined according to International Classification of Diseases diagnostic codes and Anatomical Therapeutic Classification codes for medication prescriptions. Fitting Cox hazards models, the risk of developing all-cause dementia in individuals with and without UC was estimated. Further, we compared the risk of all-cause dementia among those with and without a colectomy.

**Results** Among 4 821 488 individuals (52.6% females) followed for 84.1 million person-years between 1964 and 2018, the incidence rate of all-cause dementia was 63.90 (63.73–64.07) events per 10 000 person-years in individuals without UC, 94.80 (92.04–97.64) among those with UC, 95.01 (92.25–97.86) in those with UC but without colectomy and 63.42 (40.92–98.31) in those with UC and a colectomy. Adjusted Cox models showed an increased all-cause dementia risk in individuals with UC (HR 1.07, 95% CI 1.04 to 1.10). We found no differences between unexposed individuals and those with UC and a colectomy (HR 0.89, 95% CI 0.57 to 1.38).

**Conclusion** The findings are consistent with previous evidence suggesting a slightly increased dementia risk among individuals with UC. This study provided no evidence of further risk increase of dementia among those who had a colectomy.

## STRENGTHS AND LIMITATIONS OF THIS STUDY

⇒ This study was based on data from several Swedish registers with over 4.8 million individuals including 84.1 million person-years over seven decades, followed for an average of 17 years of observation time.
⇒ Prospectively recorded coverage from inpatient and outpatient services with a 79%–90% positive predictive value for ulcerative colitis (UC).
⇒ By using national registers, misclassification of dementia and ulcerative colitis is plausible.
⇒ Due to the limited number of individuals with UC who underwent colectomy, we could not evaluate the significance of age at colectomy and whether this association differs according to the type of dementia.

## INTRODUCTION

Inflammatory bowel disease (IBD) is a complex, multifactorial disorder with a multifaceted pathogenesis and is believed to result from inappropriate and sustained activation of the mucosal immune system that results in chronic gut inflammation. IBD is a collective term for related intestinal disorders, particularly Crohn's disease (CD) and ulcerative colitis (UC). While CD is characterised by inflammation in any part of the gastrointestinal tract and extraintestinal manifestations, UC is confined to the inner lining of the colon and rectum. Total colectomy is a major bowel surgery used in the treatment of severe and extensive UC that involves the removal of the entire colon and, often, the rectum. This surgical procedure is usually performed among those who do not respond to pharmaceutical treatment or when all the diseased tissues cannot be removed by less extensive surgery such as partial colectomy or hemicolectomy. In addition to providing relief from the disease itself, colectomy can help eliminate the cause of inflammation in the digestive tract.

The association between chronic systemic inflammation and many common psychiatric and physical conditions is well documented.[1–4] High levels of circulating proinflammatory cytokines increase the permeability of the blood–brain barrier,[5 6] thus affecting neurotransmitter systems and altering the metabolism of the central nervous system.[7 8] For instance, inflammatory cytokines (eg, TNF-α, IL-1 and IL-6) have

been associated with several mental conditions including major depressive disorder,[9] suicidal ideation,[10] vascular dementia,[11] and Alzheimer's disease.[11 12]

Therefore, it is conceivable that proinflammatory changes in the brain due to UC may mediate some of these effects and be involved in the pathophysiology of dementia. Indeed, prior research indicated an increased risk for developing dementia in those with IBD.[13 14] Despite colectomy indicating more aggressive UC behaviour, we hypothesised that colectomy might confer some protection against developing dementia in individuals with UC by eliminating chronic colonic inflammation.

This study used national Swedish register data to (1) assess the risk of developing dementia in individuals with and without UC and (2) compare this risk in those with UC who underwent colectomy with those who did not.

## METHODS
### Study design and population
This study was based on several Swedish national registers linked through the personal identification number issued to all residents in Sweden. The Total Population Register includes demographic information such as date of birth, sex, date of death and migration for all individuals born since 1932 and alive in 1963 and beyond.[15] The National Patient Register (NPR) records primary and secondary diagnoses from inpatient care since 1964 and outpatient information since 2001.[16] The Cause of Death Register (CDR) contains information on all deaths since 1952.[17] In both the NPR and CDR, diagnoses were classified according to the International Classification of Diseases (ICD) versions 7/8/9/10. Additionally, the Prescribed Drug Register (PDR) provided data on all dispensed medication prescriptions since 1 July 2005, including prescription date and dosage, using the Anatomical Therapeutic Classification system. Lastly, the Longitudinal Integration Database for Health Insurance and Labour Market Studies register, which had been compiled annually from 1990, and the Population and Housing Census (conducted every 5 years between 1960 and 1990), were used to obtain socioeconomic information.

To ensure that individuals included in the study had reached an age relevant for assessing dementia, we included all individuals born in Sweden who were at least 59 years between 1964 and 2018. The follow-up period to detect dementia in each individual began at age 59 years or the start of the NPR on 1 January 1964, whichever came later. We chose age 59 as the starting point because it is an age range where dementia becomes more prevalent in the population, although UC and colectomy status were also identified before this age. The follow-up extended until the occurrence of any of the following events: death, emigration, a diagnosis of dementia or the end of the study period on 31 December 2018, whichever happened first. Individuals with less than 1 year of follow-up were excluded.

### Patient and public involvement
Patients and/or public were not involved in the design, or conduct, or reporting, or dissemination plans of the research.

### Exposures
In this study, we included two different exposures: the occurrence of UC, with or without colectomy. Following prior research,[18] the date of diagnosis of UC was identified using ICD codes in the NPR: ICD-7: 572, ICD-8 563, 569, ICD-9 556 and ICD-10 K51. Those who had a CD diagnosis first, but followed by two or more UC diagnoses, were classified as having UC from the first diagnosis. As we only had access to three-digit codes for ICD-8, differentiating between UC and CD relied on subsequent diagnoses in the same patients using ICD-9/10 codes. Individuals who were solely identified through ICD-8 codes were excluded from the analysis (n=251). Furthermore, using data until the end of follow-up, individuals were excluded if they had an initial diagnosis of UC but subsequently received at least two CD diagnoses (n=10 682) or had a CD diagnosis not followed by at least two UC diagnoses (n=19 496; figure 1 visually depicts the process of identifying the individuals included in the analysis). Among individuals with UC, we identified individuals who underwent total colectomy and were identified using the Swedish Classification of Operation and Major Procedures codes: JFH00, JFH01, JFH10, JFH11, JFH20, JFH30, JFH33, JFH40 and JFH96. Both exposures were allowed to vary over time, that is, individuals started as non-exposed until the occurrence of each exposure separately and were considered as exposed from that time onwards. Those who had UC or a colectomy before the start of follow-up were coded as exposed from the start of follow-up. We created two variables: one indicating the presence or absence of a UC diagnosis (1/0) and a second variable further breaking down people with UC into with and without colectomy and compared both groups to those without UC.

### Outcome
All-cause dementia was defined as either (1) a recorded diagnosis in the NPR using ICD codes: ICD-7: 304, 305 and 306, ICD-8 290, 293.0–293.1, ICD-9 290A, 290B, 290E, 290W, 290X, 294B, 331A–C, 331X and ICD-10 F00–F01, F02.3, F03, F05.1, F09, G30 or G31.1, (2) at least two recorded dispensed medications of any of the following anticholinesterases: N06DA01, N06DA03, N06DA04, N06D×01, N06DA02, N06DA52 or (3) a recorded entry in the Death register with the previous all-cause dementia ICD codes. Alzheimer's disease included ICD-10 codes F00, G30, ICD-9 codes 290A/B, 331A and ICD-8 code 290, ICD-7 code 305.00. Vascular dementia included ICD-10 codes F01, ICD-9 290E, and ICD-8 codes 293.0–293.1, and ICD-7 code 306.99. Given the insidious nature of dementia, the date of the onset of all-cause dementia was defined as 3 years before the diagnosis of dementia in the NPR or a dispensed medication and 5 years before the death. Individuals with a diagnosis of dementia before

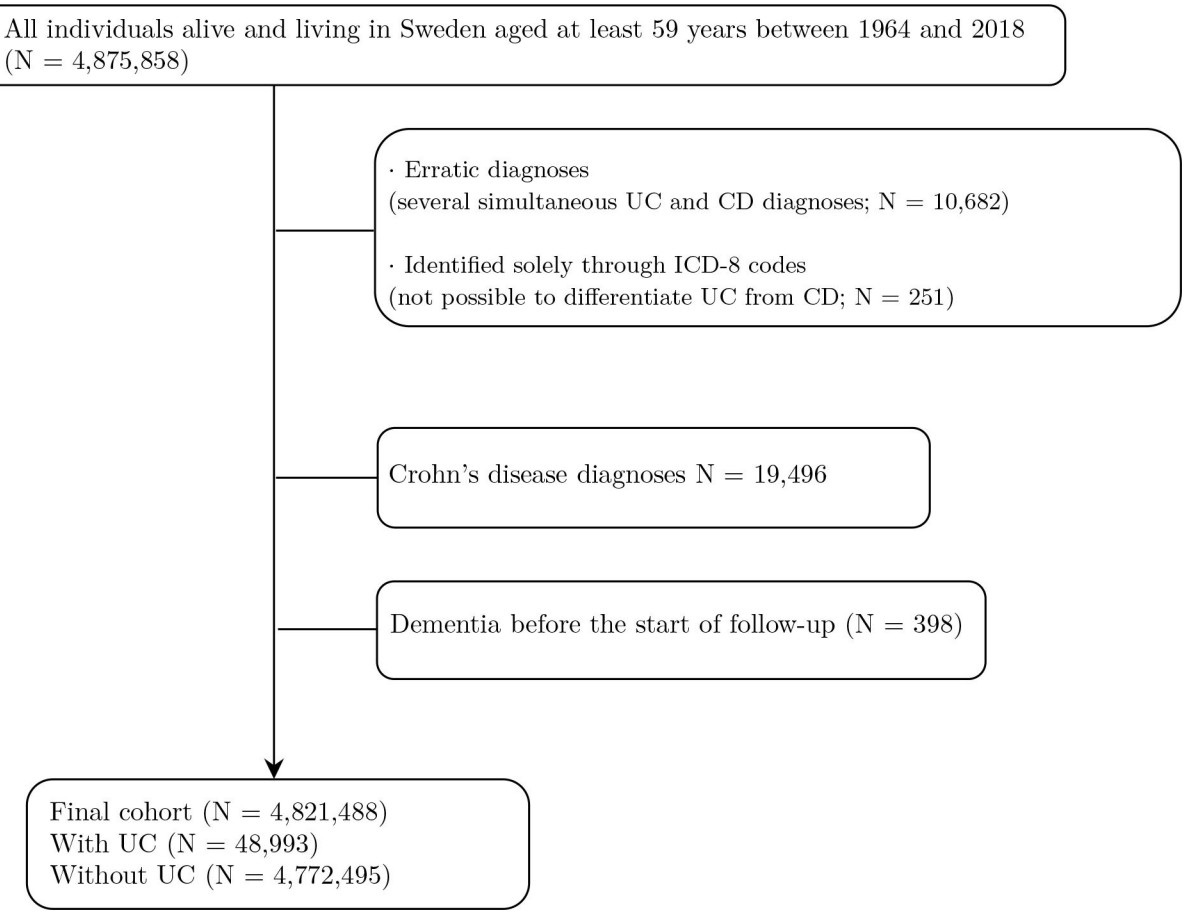

All individuals alive and living in Sweden aged at least 59 years between 1964 and 2018
(N = 4,875,858)

· Erratic diagnoses
(several simultaneous UC and CD diagnoses; N = 10,682)

· Identified solely through ICD-8 codes
(not possible to differentiate UC from CD; N = 251)

Crohn's disease diagnoses N = 19,496

Dementia before the start of follow-up (N = 398)

Final cohort (N = 4,821,488)
With UC (N = 48,993)
Without UC (N = 4,772,495)

**Figure 1** Flow chart illustrating the screening and inclusion process of study participants based on the inclusion and exclusion criteria. CD, Crohn's disease; ICD, International Classification of Disease; UC, ulcerative colitis.

the start of follow-up were excluded from the analysis (n=398; figure 1).

## Covariates

Potential confounding factors included birth year, sex, region (Norrland, Svealand and Götaland), socioeconomic index (SEI) and the Household Crowding Index (HCI). Using the Population and Housing Census (Swedish acronym, FOB; 1960–1990 every 10 years), we extracted information on SEI and HCI using the year closest to birth (when not available, information was taken from the census up to age 10 years). SEI was based on the occupation of the father of the household (or the mother when there was no information on the father). The HCI was calculated as the ratio of total number of individuals in the household to the number of rooms in the household.

## Statistical analysis

Demographic and other individual baseline characteristics were summarised by median, proportions and IQR for individuals without UC and for those with UC who did and did not undergo colectomy. Incidence rates (IRs) of dementia with 95% CIs were calculated fitting a Poisson generalised linear model with the log link function and robust SEs. A Cox proportional

hazards model with age as the underlying time scale[19] was fitted to estimate HRs with 95% CIs for the association between colectomy and dementia onset. Both exposures (UC and total colectomy) were modelled as time-dependent covariates, classifying individuals as exposed only after the occurrence of each exposure. Analyses were adjusted for sex, birth year and county of birth. First, we examined the association between UC (regardless of colectomy) and the risk of developing all-cause dementia. Subsequently, we separated the UC group into two subgroups based on colectomy status: those with and without a colectomy. By conducting separate analyses for each subgroup, we were able to assess associations with dementia risk for UC with and without colectomy. Shoenfeld residuals were inspected to assess the proportionality assumption and when violated, analyses were stratified by the variables that did not meet this assumption. Data management and statistical analyses were performed using R V.4.2.2.[20]

## Sensitivity analysis

Due to incomplete coverage in the NPR, the analysis was repeated including only individuals who were 60 years old after 1973 to ensure greater coverage.

**Table 1** Characteristics of the cohort (N=4 821 488)

| Variable | Without IBD (N=4 772 495)* | UC without colectomy† (N=48 500)* | UC with colectomy† (N=493)* |
|---|---|---|---|
| Median follow-up time (years) | 17 (10, 25) | 17 (10, 24) | 13 (7, 19) |
| Age at the start of follow-up (years) | 59 (59, 59) | 59 (59, 59) | 59 (59, 59) |
| Sex (%) | | | |
| Male | 2 263 178 (47) | 23 513 (48) | 297 (60) |
| Female | 2 509 317 (53) | 24 987 (52) | 196 (40) |
| Household Crowding Index (HCI) | 1.50 (1.00, 2.00) | 1.50 (1.00, 2.00) | 1.50 (1.20, 2.00) |
| Missing | 2 917 369 | 27 917 | 186 |
| Socio-Economic Index (SEI; %) | | | |
| Agriculture | 289 162 (6.1) | 2991 (6.2) | 43 (8.7) |
| High | 212 356 (4.4) | 2315 (4.8) | 35 (7.1) |
| Low | 801 437 (17) | 9268 (19) | 138 (28) |
| Medium | 463 642 (9.7) | 4972 (10) | 73 (15) |
| Unknown | 2 918 081 (61) | 27 919 (58) | 186 (38) |
| Other | 87 817 (1.8) | 1035 (2.1) | 18 (3.7) |
| County (%) | | | |
| Götaland | 2 126 607 (45) | 22 223 (46) | 190 (39) |
| Norrland | 679 216 (14) | 7004 (14) | 52 (11) |
| Svealand | 1 657 347 (35) | 17 397 (36) | 232 (47) |
| Unknown | 309 325 (6.5) | 1876 (3.9) | 19 (3.9) |
| Median age of colectomy (years) | 73 (67, 79) | 0 | 65 (56, 72) |
| Without colectomy | 4 771 206 | 48 500 | 0 |
| All-cause dementia (%) | 541 574 (11) | 4785 (9.9) | 28 (5.7) |
| Alzheimer's disease | 146 689 (3.1) | 1261 (2.6) | 7 (1.4) |
| Vascular dementia | 70 615 (1.5) | 652 (1.3) | 2 (0.4) |

*Median (IQR); n (%).
†Variables defined as present from the start of follow-up.
IBD, inflammatory bowel disease; UC, ulcerative colitis.

## RESULTS

This cohort study comprised 4 821 488 individuals with a median (IQR) age at the start of follow-up of 59 (SD=0.71) years, of whom 2 534 500 (52.6%) were female and 2 286 988 (47.4%) were male. Among those, 48 993 (1%) had a diagnosis of UC, of whom 493 (1%) had a colectomy before or during follow-up, with a median (IQR) age at the operation of 63 (51–74) years. During a total follow-up time of 84.1 million person-years between 1964 and 2018, a diagnosis of all-cause dementia was observed in 546 387 (11.3%) individuals. A summary of the cohort's demographic characteristics is presented in table 1.

The IR of all-cause dementia was 63.90 (63.73–64.07) events per 10 000 person-years in individuals without UC 94.80 (92.04–97.64) among those with UC 95.01 (92.25–97.86) in those with UC but without colectomy and 63.42 (40.92–98.31) for those with UC and a colectomy. Cox models adjusted for sex, birth year and birth county showed that individuals with UC had an increased all-cause dementia risk compared with the general population (HR 1.07, 95% CI 1.04 to 1.10). When we separated individuals with UC into with or without a colectomy and compared both groups to individuals without UC, dementia risk for individuals with UC but without colectomy increased (HR 1.07, 95% CI 1.04 to 1.10), but the risk became non-significant for individuals with UC and a total colectomy (HR 0.89, 95% CI 0.57 to 1.38). Individuals with UC who underwent a colectomy showed a suggestive decrease in the risk of dementia (HR 0.83, 95% CI 0.54 to 1.30) compared with those with UC but without a colectomy; however, the wide CI indicated substantial uncertainty.

Sensitivity analyses showed that after restricting analyses to those who started follow-up in 1974 and onwards, the magnitude of the association decreased compared with the main analysis but still indicated that those with UC were at increased risk for all-cause dementia (HR 1.04, 95% CI 1.01 to 1.08). For those with UC and without a colectomy, we observed an increased risk for

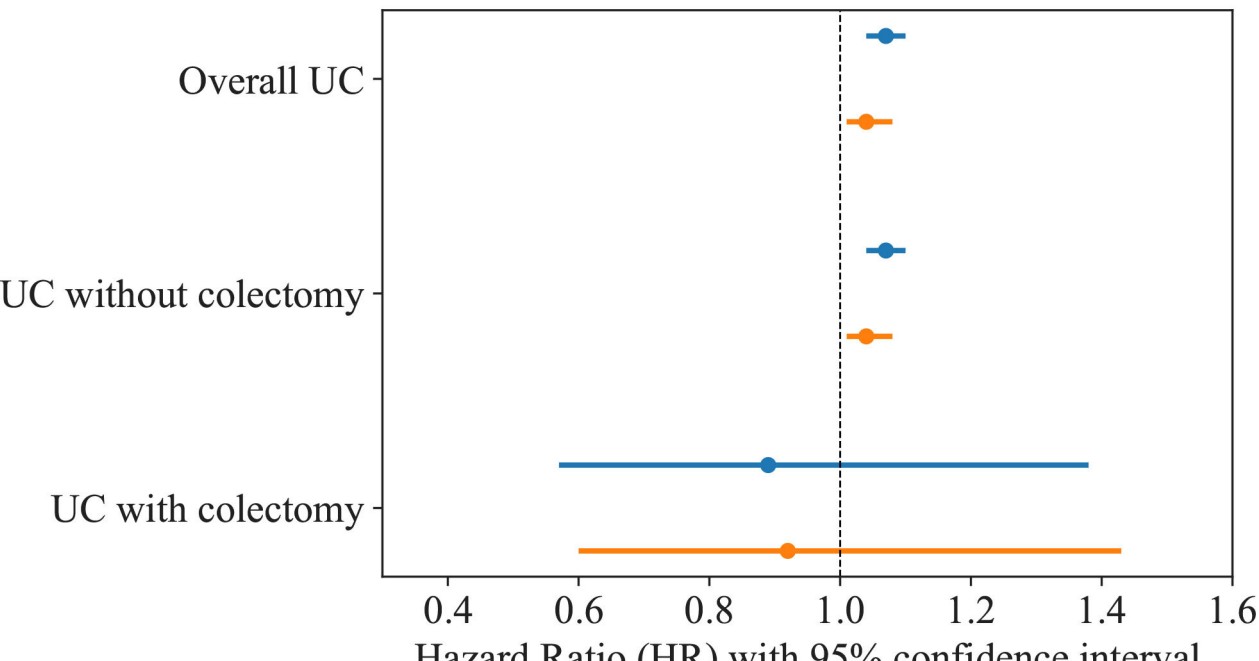

**Figure 2** Forest plot of the association between UC and all-cause dementia in those with UC (with and without total colectomy) compared with unexposed individuals. UC, ulcerative colitis.

all-cause dementia compared with the general cohort (HR 1.04, 95% CI 1.01 to 1.08) and no difference in the hazard between those with UC and a total colectomy and the general population (HR 0.92, 95% CI 0.60 to 1.43). Figure 2 summarises the results of the main and sensitivity analyses.

## DISCUSSION

This large cohort study of individuals from the general population with prospectively recorded data assessed the association between UC and subsequent development of all-cause dementia and investigated the previously unexplored topic, to the best of our knowledge of the potential role of colectomy in this relationship. The results indicated that individuals with UC are at slightly increased all-cause dementia risk compared with those without UC, but there was no evidence of an even greater magnitude risk in those who have a colectomy.

Recent meta-analysis evidence indicated an increased dementia risk in individuals with IBD with a pooled risk ratio of 1.35 (1.08–1.68).[14] Our findings are consistent with this finding and mirror conclusions obtained in previous studies suggesting a relationship between UC and dementia, with a similar magnitude of the association to those obtained in Danish register studies[13] and Canadian[21] and German cohort studies.[22] Our study provides novel findings by indicating that having a total colectomy does not further increase dementia among people

with UC. While our results are consistent with a possible reduction of dementia risk in UC following colectomy, the interpretation remains equivocal. It is noteworthy that as colectomy signals more severe UC disease activity, as well as the inconvenience postcolectomy, there was no additional dementia risk compared with those with UC without colectomy.

While the precise aetiology of UC remains elusive, it has been proposed that a dysregulated mucosal immune response, in the context of a disturbed gut microbiota, plays a pivotal role in the onset and progression of the disease.[23–26] The intestinal immune response is a process composed of complex interactions between several cell types interacting within a specific microenvironment, the intestinal mucosa. When these interactions are perturbed by dysbiosis, such as during the onset of IBD, an excessive immune response occurs, with an aberrant mucosal immune response that involves the production of large amounts of proinflammatory cytokines, such as interleukins, which can lead to severe tissue damage[27] and it has been associated with several psychiatric conditions, including dementia.[1–4 9–12] A balance between proinflammatory and anti-inflammatory signals is necessary for the host to avoid acute or chronic inflammation and its harmful consequences. We hypothesise that with the removal of the colon, the excessive immune response and the overproduction of proinflammatory cytokines cease, reducing colonic and potentially chronic systemic

inflammation and, in turn, potentially reducing the risk of dementia.

## Strengths and limitations

This study presents several noteworthy strengths, including a nationwide sample of over 4.8 million individuals including 84.1 million person-years over seven decades followed for an average of 17 years of observation time with prospectively recorded coverage from inpatient and outpatient services with a 79%–90% positive predictive value for UC.[28] To further reduce UC misclassification, we required patients to have at least two consecutive UC diagnoses. There are potential limitations that should be considered. First, the estimated increased risk for dementia in people with UC may be, in part, explained by surveillance bias, where people with UC have increased contact with healthcare systems. However, those with UC and a colectomy also experience closer contact with healthcare services and we did not observe an increased risk. Second, although we obtained all-cause dementia diagnoses from inpatient and outpatient care services, misclassification is possible. To reduce misclassification, we retrieved all prescribed anticholinesterases used to treat dementia from the PDR from 2005 onwards (>99% coverage). However, diagnoses of dementia and UC before 2001 were based solely on inpatient data. Without outpatient data prior to 2001, there is a possibility of underidentification of dementia and UC diagnoses. It is crucial to interpret the results within the context of these data constraints and acknowledge the potential biases that may arise due to incomplete data capture before the specified time periods. Nonetheless, our estimated IR (6.4 per 1000 person-year) is smaller than those recently reported in the literature,[29] highlighting potential misclassification. This misclassification—most likely non-differential—refers to individuals with dementia that are not identified leading to systematic error in the estimation of association and would bias results towards the null. Further, given the low count of individuals with UC who had a colectomy, we were not able to assess the importance of age at colectomy and whether this association varies by dementia type.

## Conclusions

Our findings contribute to the growing body of literature on the increased risk of developing dementia among individuals with UC. Additionally, our study indicated that total colectomy in UC patients is not associated with an even higher magnitude of increased dementia risk.

**Contributors** MG-A had full access to all the data in the study and took responsibility for the integrity of the data and the accuracy of the data analysis. Concept and design: all authors. Statistical analysis: MG-A. Acquisition, analysis or interpretation of data: all authors. Drafting of the manuscript: MG-A. Critical revision of the manuscript for important intellectual content: all authors. Supervision: SM and AH. MG-A is guarantor for the paper.

**Funding** This work was supported by grant from the Swedish Research Council for Health, Working Life and Welfare (number: 2019-01236).

**Competing interests** None declared.

**Patient and public involvement** Patients and/or the public were not involved in the design, or conduct, or reporting, or dissemination plans of this research.

**Patient consent for publication** Not applicable.

**Ethics approval** The study had ethical approval from the Swedish Ethical Review Authority (Dnr 2019-04755, 2020-02406 and 2022-00336-02). Requirement for informed consent was waived for the current study because it was a secondary analysis of existing data. The investigation conforms to the 1964 Declaration of Helsinki and its later amendments or comparable ethical standards.

**Provenance and peer review** Not commissioned; externally peer reviewed.

**Data availability statement** Data may be obtained from a third party and are not publicly available. The Public Access to Information and Secrecy Act in Sweden prohibits us from making individual level data publicly available. Researchers who are interested in replicating our work can apply for individual level data at Statistics Sweden: www.scb.se/en/services/guidance-for-researchers-and-universities/.

**ORCID iDs**
Miguel Garcia-Argibay http://orcid.org/0000-0002-4811-2330
Scott Montgomery http://orcid.org/0000-0001-6328-5494

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
