## [Reviewer comments · BMJ Open]

ARTICLE DETAILS

TITLE (PROVISIONAL)	Association between dementia risk and ulcerative colitis, with and without colectomy: a Swedish population-based register study
AUTHORS	Garcia-Argibay, Miguel; Hiyoshi, Ayako; Montgomery, Scott

VERSION 1 – REVIEW

REVIEWER	Qi, Xingshun General Hospital of Shenyang Military Region
REVIEW RETURNED	07-May-2023

GENERAL COMMENTS	General comments to the Authors The title “Dementia risk and ulcerative colitis, with and without colectomy: a Swedish population-based register study.” is a bit confusing. It can be revised as the “association between dementia risk and UC with and without colectomy ...”. Abstract In the sentence “to investigate the association of ulcerative colitis.....”, “to” should be revised as “To”. The words “..... and assess differences in those with a total colectomy” may be revised as “..... and assess differences in those with and without a total colectomy”. In the Primary and secondary outcome measures section and the Keyword section, the “:” should be deleted after the subtitles. In the Results section, the authors said “..... 95.01 (92.25-97.86) in those with UC but without colectomy, and 63.42 (40.92-98.31) in those with UC and a colectomy.” But they said “95.01 (92.25-98.31) in those with UC but without colectomy, and 63.42 (40.92-98.31) for those without UC.”. The authors should check the data carefully. The sentence “those with UC and a colectomy showed no increased risk (HR=0.89 [0.57-1.38])” is a bit confusing, because the “HR=0.89 [0.57-1.38]” represents that the dementia risk is not statistically significant for individuals with UC and a total colectomy (increased or decreased?). Methods The authors should consider whether the patients used drugs for the treatment of UC during the follow-up period and whether these drugs have affected dementia risk. For example, whether the use of the related drugs in patients with UC but without a colectomy affects the risk of dementia. The authors should draw a flow chart for screening patients according to the inclusion and exclusion criteria. Results
---

	In the full-text, the authors only list the incidence rate of all-cause dementia in patients with UC but without colectomy and in patients with UC and a colectomy. But the authors also need to list the incidence rate of all-cause dementia in people with and without UC.
--	---

REVIEWER	Troelsen, Frederikke Aarhus Universitet
REVIEW RETURNED	08-May-2023

GENERAL COMMENTS	Thank you for giving me the opportunity to review the manuscript entitled: "Dementia risk and ulcerative colitis, with and without colectomy: a Swedish population-based register study". The study deals with an interesting and relevant issue but has some important limitations that I think need to be considered and revised carefully. These limitations include an insufficient description of the defined follow-up period, analyses, and results as well as an insufficient number of stratified analyses (for baseline characteristics, time periods, and subtypes of dementia). In addition, the study results would, to a greater extent, be applicable to clinical practice if cumulative incidence proportions of all-cause dementia were calculated and included alongside the incidence rates. Based on my previous experience with registry-based research in Scandinavia, these limitations are correctable, and the manuscript will thus benefit substantially from a thorough revision. Please find my comments to the authors attached. Comments to the authors: Summary: Based on Swedish registry data, the authors aimed to investigate the risk of all-cause dementia among individuals with and without ulcerative colitis (UC). The authors also investigated the impact of prior colectomy for UC on risk of all-cause dementia comparing patients with UC and colectomy to the general population. The main findings of the study were a slightly increased risk of all-cause dementia among UC patients compared with the general population (HR: 1.07 [95% CI: 1.04-1.10]) and a slightly decreased risk of all-cause dementia among UC patients with a colectomy compared with the general population (HR: 0.89 [95% CI: 0.57-1.38]). The study deals with a relevant clinical issue, but I do have some comments that need to be addressed. Please find my comments to consider below: Major comments:  1. The manuscript in its current form lacks a clear description of the follow-up period. I have the following comments and suggestions to consider:  a. Page 7, line 6 reads: "Individuals were followed from age 59 years or the beginning of the NPR in 1964". It is not clear how this statement fits together with the statement on page 10, line 10 "During a total follow-up period of 84.1 million person-years between 1939 and 2018"? How is it possible to include person-years before 1964 when the NPR started recording of inpatients? b. It is not entirely clear why the patients were followed from age 59 and not the diagnosis of UC? A clearer description of when the individuals enter the cohort and when the individuals contribute with person-time is needed. c. Why do the authors include only those who were born before or in 1958? Further clarifications would be helpful. d. How many individuals without UC developed UC during follow-up and were these included as exposed?
--

	e. The revised description of follow-up may be moved to the statistical analyses part of the manuscript. 2. Data availability changed during the study period. For instance, outpatient data from the NPR were available from 2001 and medication data were available from 2005. The manuscript may benefit from including stratified analyses on relevant time periods to examine the impact of changing data availability over the study period. This is somehow addressed in the sensitivity analysis but could be even more sufficiently investigated. 3. A previous study showed variations in risk estimates for different types of dementia. Based on the included data in the present study it should be possible to conduct analyses stratified by subtypes of dementia (i.e., Alzheimer’s Disease, vascular dementia, frontotemporal dementia etc.). 4. The authors investigated the risk of all-cause dementia among UC patients with colectomy compared with the risk of all-cause dementia in the general population. Did the authors consider investigating the impact of colectomy on all-cause dementia risk in UC patients? (e.g., to compare the risk of all-cause dementia in UC patients with colectomy to UC patients without colectomy). 5. The results presented in the abstract and results sections describe the incidence rate of all-cause dementia for the complete cohort (i.e., for UC and non-UC patients combined). Please consider separating the results and provide estimates for individuals without UC, with UC, with UC and colectomy, and with UC and without colectomy. 6. Did the authors consider including cumulative incidence proportions (treating death as a competing risk) of dementia for individuals without UC, with UC, with UC and colectomy, and with UC and without colectomy alongside the incidence rates? Cumulative incidence proportions could be more easily applicable to clinical practice. 7. The manuscript would benefit from adding an analysis on all-cause dementia stratified by baseline characteristics. 8. Although results were not statistically significant, they indicated a decreased risk of all-cause dementia among UC patients who underwent a colectomy. This finding could be emphasized in the discussion part. Minor comments: 1. Page 4, line 8: The 86.7 million person-years seems wrong as the abstract (page 3, line 26) states that individuals were followed for 84.1 million person-years? 2. Please include in the description of the LISA which period this registry covers. 3. The definitions and data concerning the outcome are described before details concerning the exposure. The authors could consider describing the exposure before the outcome as the present study is a cohort study. 4. It is described under the exposure part how individuals were treated if they received both UC and Crohn’s Disease diagnoses. This section might benefit from a graphical illustration of different scenarios. 5. Page 9, line 14: Please consider changing the term “overall cohort” to “without UC” or similar. 6. Page 9, line 28: The cox-proportional hazards regression analyses were used for investigating the association between colectomy and development of dementia. What about the association between UC combined and risk of dementia?
--	--

VERSION 1 – AUTHOR RESPONSE

Reviewer: 1

Comment 1: The title “Dementia risk and ulcerative colitis, with and without colectomy: a Swedish population-based register study.” is a bit confusing. It can be revised as the “association between dementia risk and UC with and without colectomy ...”.

Answer: We thank the Reviewer for this suggestion. The title has been changed accordingly.

Comment 2: In the sentence “to investigate the association of ulcerative colitis.....”, “to” should be revised as “To”.

Answer: We thank the Reviewer for noticing this type. We have now corrected it.

Comment 3: The words “..... and assess differences in those with a total colectomy” may be revised as “..... and assess differences in those with and without a total colectomy”.

Answer: The sentence has been corrected and now states: “differences in those with and without a total colectomy.”

Comment 4: In the Primary and secondary outcome measures section and the Keyword section, the “:” should be deleted after the subtitles.

Answer: The colon has been removed from the “primary and secondary outcome measures” and “keyword”. We thank the Reviewer for noticing it.

Comment 5: In the Results section, the authors said “..... 95.01 (92.25-97.86) in those with UC but without colectomy, and 63.42 (40.92-98.31) in those with UC and a colectomy.” But they said “95.01 (92.25-98.31) in those with UC but without colectomy, and 63.42 (40.92-98.31) for those without UC.”. The authors should check the data carefully.

Answer: We thank the Reviewer for spotting this mistake in the results section. Indeed, 63.42 (40.92-98.31) is for those with UC and a total colectomy. This was correct in the abstract and has been corrected. The text now states:

“...95.01 (92.25–97.86) in those with UC but without colectomy, and 63.42 (40.92–98.31) for those with UC and a colectomy.”

Comment 6: The sentence “those with UC and a colectomy showed no increased risk (HR=0.89 [0.57-1.38])” is a bit confusing, because the “HR=0.89 [0.57-1.38]” represents that the dementia risk is not statistically significant for individuals with UC and a total colectomy (increased or decreased?).

Answer: We thank the Reviewer for bringing up this concern. The hazard ratio (HR=0.89 [0.57-1.38]) indicated that there was no statistically significant difference in dementia risk for individuals with UC and a total colectomy relative to the comparison group.

Our initial hypothesis was that having a colectomy could potentially provide some level of protection against dementia. However, due to the uncertainty associated with the hazard ratio estimate (CI 0.57-1.38), we cannot draw definitive conclusions regarding the potential protective effect of colectomy against dementia in individuals with UC. Indeed, the point estimate of 0.89 suggests a slightly decreased risk of dementia among those with UC and a colectomy, but the wide confidence interval

reflects the uncertainty and imprecision surrounding the estimate. Therefore, it would be inappropriate to place excessive emphasis on the non-significant estimate given its uncertainty. As such, we reported that we could not find evidence to support any significant difference in dementia risk compared to unexposed individuals.

We have changed that sentence to reflect the uncertainty.

“We found no differences between unexposed individuals and those with UC and a colectomy (HR=0.89 [0.57-1.38]).”

Methods

Comment 7: The authors should consider whether the patients used drugs for the treatment of UC during the follow-up period and whether these drugs have affected dementia risk. For example, whether the use of the related drugs in patients with UC but without a colectomy affects the risk of dementia.

Answer: We thank the Reviewer for this suggestion. We chose not to adjust for drugs related to the treatment of UC in our analysis for two reasons. Firstly, adjusting for these drugs may inadvertently remove the mediating effect of the drugs, introducing bias in estimating the direct effect of UC on dementia due to very plausible mediator-outcome confounding. Secondly, our objective was to estimate the total effect of UC on dementia, and adjusting for mediators would hinder the assessment of the total effect. This reflects advice from the book "Statistical Rethinking" by McElreath, which provides detailed insights into this topic.

Comment 8: The authors should draw a flow chart for screening patients according to the inclusion and exclusion criteria.

Answer: We have incorporated a flowchart into the revised manuscript. The flowchart serves to illustrate the step-by-step process of patient screening, clearly outlining the inclusion and exclusion criteria.

Results

Comment 9: In the full-text, the authors only list the incidence rate of all-cause dementia in patients with UC but without colectomy and in patients with UC and a colectomy. But the authors also need to list the incidence rate of all-cause dementia in people with and without UC.

Answer: We thank the Reviewer for this suggestion. We have now included the incidence rate of all-cause dementia in patients with UC, IR=**94.80 (92.04,97.64)**. It now reads:

“The incidence rate of all-cause dementia was 63.90 (63.73–64.07) events per 10,000 person-years in individuals without UC, 94.80 (92.04–97.64) among those with UC, 95.01 (92.25–97.86) in those with UC but without colectomy...”

Reviewer: 2

Major comments:

1. The manuscript in its current form lacks a clear description of the follow-up period. I have the following comments and suggestions to consider:

a. Page 7, line 6 reads: “Individuals were followed from age 59 years or the beginning of the NPR in 1964”. It is not clear how this statement fits together with the statement on page 10, line 10 “During a total follow-up period of 84.1 million person-years between 1939 and 2018”? How is it possible to include person-years before 1964 when the NPR started recording of inpatients?

Answer: We thank the Reviewer for finding this typo. Indeed, the lower bound was wrong and it has been corrected: 1964–2018.

b. It is not entirely clear why the patients were followed from age 59 and not the diagnosis of UC? A clearer description of when the individuals enter the cohort and when the individuals contribute with person-time is needed.

Answer: We appreciate the reviewer's comment regarding the starting point of follow-up in our study. Since not all individuals had UC, we needed to set an age to start follow-up. The decision to begin following individuals from age 59 was driven by several considerations. Firstly, age 59 was chosen as it aligns with an age range where dementia becomes more prevalent in the population. By starting at this age, we aimed to capture a meaningful period for assessing the association between UC and dementia. Importantly, however, while follow-up for dementia began at age 59 years, the study identified UC and colectomy prior to this age.

Additionally, starting at a specific age, such as 59, allows for a consistent baseline age across all participants in the study. This helps to reduce potential confounding effects related to age, as individuals within the cohort are comparable in terms of their baseline characteristics. Furthermore, by employing a time-varying approach, we were able to model UC diagnosis during the study period appropriately. This allowed us to include their non-exposed time in the analysis, providing a comprehensive assessment of the association between UC and dementia.

c. Why do the authors include only those who were born before or in 1958? Further clarifications would be helpful.

Answer: We included individuals born after 1958 because given the follow-up was from age 59, the study population needed to reach age of 59 by the end of the study period in 2018.

d. How many individuals without UC developed UC during follow-up and were these included as exposed?

Answer: As stated in the methods, the study utilized a time-varying approach, which means that individuals could transition from being nonexposed to being exposed to UC during the follow-up period. Therefore, individuals who had UC by the start of follow-up were classified into the exposed category from the start of follow-up, whereas individuals without UC at the start of follow-up but developed it during the follow-up were modelled as nonexposed until they develop UC and as exposed from the date of UC diagnosis.

e. The revised description of follow-up may be moved to the statistical analyses part of the manuscript.

Answer:

We have rephrased the whole paragraph explaining follow-up based on previous suggestions. The manuscript now states:

“In order to ensure that individuals included in the study had reached an age relevant for assessing dementia, we included all individuals born in Sweden who were at least 59 years between 1964 and 2018. The follow-up period to detect dementia in each individual began at age 59 years or the start of the NPR on January 1, 1964, whichever came later. We chose age 59 as the starting point because it is an age range where dementia becomes more prevalent in the population, although UC and colectomy status were also identified before this age. The follow-up extended until the occurrence of any of the following events: death, emigration, a diagnosis of dementia, or the end of the study period on December 31, 2018, whichever happened first.”

Regarding the placement of the revised description of follow-up, we have decided to keep it under the "study design and population" section rather than within the statistical analyses. This is because the description provides important information about the cohort and follow-up times, which are fundamental aspects of the study design, rather than specific statistical methods or analyses. We believe this placement helps provide a comprehensive understanding of the study design and population.

2. Data availability changed during the study period. For instance, outpatient data from the NPR were available from 2001 and medication data were available from 2005. The manuscript may benefit from including stratified analyses on relevant time periods to examine the impact of changing data availability over the study period. This is somehow addressed in the sensitivity analysis but could be even more sufficiently investigated.

Answer: We Thank the Reviewer for this suggestion. We recognize the importance of examining the impact of changing data availability over the study period. Indeed, for this purpose, we conducted a sensitivity analysis by including individuals who were 60 years old after 1973 because the coverage of the inpatient register was expanded greatly from around 1970. However, it is important to note that further restricting the start of follow-up would result in low cell counts and thus, increased imprecision, particularly in the group of individuals with UC and a colectomy. While we understand the value of conducting stratified analyses on relevant time periods, the limited sample size in our study would have rendered such analyses unstable and potentially misleading.

We have further expanded the limitation section to include this aspect. It now reads:

“However, diagnoses of dementia and UC before 2001 were based solely on inpatient data. Without outpatient data prior to 2001, there is a possibility of under-identification of dementia and UC diagnoses. It is crucial to interpret the results within the context of these data constraints and acknowledge the potential biases that may arise due to incomplete data capture before the specified time periods.(P12).

3. A previous study showed variations in risk estimates for different types of dementia. Based on the included data in the present study it should be possible to conduct analyses stratified by subtypes of dementia (i.e., Alzheimer’s Disease, vascular dementia, frontotemporal dementia etc.).

Answer: In our cohort, we only observed two individuals with vascular dementia among those with UC and colectomy, and seven with Alzheimer's disease. These numbers have been now included in Table 1. While we agree that the analysis focusing on dementia type is of interest, given the number of events, we consider that results will be difficult to interpret. The limitation of small numbers is addressed in the discussion and future studies should try to replicate our findings and attempt to examine if this association varies by dementia type.

4. The authors investigated the risk of all-cause dementia among UC patients with colectomy compared with the risk of all-cause dementia in the general population. Did the authors consider investigating the impact of colectomy on all-cause dementia risk in UC patients? (e.g., to compare the risk of all-cause dementia in UC patients with colectomy to UC patients without colectomy).

Answer: We appreciate the reviewer's attention to this aspect. Indeed, we investigated the impact of colectomy on the risk of all-cause dementia among UC patients. We compared the risk of all-cause dementia in UC patients with colectomy to UC patients without colectomy. Our analysis showed a somewhat decreased risk among UC patients with colectomy, although the confidence intervals were wide, indicating substantial uncertainty around the estimate. However, considering the limited precision of this estimate and its similarity to the first analysis, we initially decided not to include it in the final analysis and reflect it in Figure 1. This comparison has been included in the revised manuscript.

"Individuals with UC who underwent a colectomy showed a suggestive decrease in the risk of dementia (HR=0.83 [0.54–1.30]) compared to those with UC but without a colectomy, however, the wide confidence interval indicated substantial uncertainty." (P10)

5. The results presented in the abstract and results sections describe the incidence rate of all-cause dementia for the complete cohort (i.e., for UC and non-UC patients combined). Please consider separating the results and provide estimates for individuals without UC, with UC, with UC and colectomy, and with UC and without colectomy.

Answer: We have now also included the incidence rate for UC in the results and in the abstract. We thank the Reviewer for highlighting this.

Results:

"The incidence rate of all-cause dementia was 63.90 (63.73–64.07) events per 10,000 person-years in individuals without UC, 94.80 (92.04–97.64) among those with UC, 95.01 (92.25–97.86) in those with UC but without colectomy, and 63.42 (40.92–98.31) for those with UC and a colectomy."

6. Did the authors consider including cumulative incidence proportions (treating death as a competing risk) of dementia for individuals without UC, with UC, with UC and colectomy, and with UC and without colectomy alongside the incidence rates? Cumulative incidence proportions could be more easily applicable to clinical practice.

Answer: We appreciate the reviewer's suggestion of including cumulative incidence proportions and competing risks in our analysis. To the best of our knowledge, there is no straightforward implementation of a sub-distribution hazards model that accommodates time-varying covariates (both UC and colectomy in our case). Regardless, the model requires considerable care, and their inclusion may have undesirable effects on the interpretability of the resultant model. 10.1002/sim.8399.

Furthermore, authors such as Paul Allison do not recommend using sub-distribution models aiming for causal inference.

Given the complexity and potential limitations associated with including time-varying covariates in competing risks analysis (e.g., Fine and Gray), we decided to focus our analysis on estimating hazard ratios using Cox proportional hazards models. This approach allowed us to investigate the association between UC, colectomy, and dementia risk while considering time-varying exposure status.

We acknowledge the importance of cumulative incidence proportions and competing risks in clinical practice. However, in the context of our study and the available methods, it was not feasible to incorporate these measures alongside time-varying covariates. We believe that the hazard ratio estimates obtained from our Cox models still provide valuable information on the association between UC, colectomy, and dementia risk in our study population. It is also worth noting that we included death with dementia (underlying cause) as our event.

7. The manuscript would benefit from adding an analysis on all-cause dementia stratified by baseline characteristics.

Answer: We appreciate the valuable suggestion from the reviewer to conduct stratified analyses on all-cause dementia based on baseline characteristics. We agree that such analyses can provide valuable insights into potential effect modification and heterogeneity of the associations. However, we would like to draw the Reviewer's attention to the limited number of individuals with dementia and UC + colectomy in our study, we have constraints on the number of stratifications that can be performed.

Given the relatively small sample size of individuals with dementia and UC plus colectomy, conducting numerous stratified analyses would result in small cell sizes within each stratum. This would lead to unstable estimates and increased uncertainty, compromising the robustness and interpretability of the findings. Our primary goal was to provide a comprehensive assessment of the overall association between UC, colectomy, and all-cause dementia, which required focusing on the main analysis.

While we understand the importance of exploring potential effect modification, we had to prioritize statistical power and the stability of our estimates. Conducting extensive stratified analyses with limited sample sizes could potentially yield unreliable and inconclusive results. However, we acknowledge that future studies with larger sample sizes specifically targeting these subgroups could provide more in-depth insights into potential effect modifiers and subgroup-specific associations.

8. Although results were not statistically significant, they indicated a decreased risk of all-cause dementia among UC patients who underwent a colectomy. This finding could be emphasized in the discussion part.

Answer: While we agree with the Reviewer about the decreased risk, it was important for us to consider the limitations associated with non-significant estimates and wide confidence intervals.

Although the point estimate suggests a decreased risk of all-cause dementia among UC patients who underwent a colectomy, the wide confidence interval indicated a considerable level of uncertainty surrounding this estimate. Placing too much emphasis on non-significant estimates, particularly when accompanied by wide confidence intervals, can lead to misleading interpretations. We attempted to interpret non-significant results cautiously and not overinterpret them as evidence of a true association or a conclusive finding.

In our discussion, we aimed to provide a comprehensive overview of the findings, including both statistically significant and non-significant results. While we acknowledge the observed trend of decreased risk among UC patients who underwent a colectomy, we must be cautious in drawing definitive conclusions due to the lack of statistical significance and the wide confidence interval. It is

important to consider these limitations and encourage further research with larger sample sizes and follow-up to obtain more precise estimates and validate these findings.

By providing a balanced discussion that acknowledges both significant and non-significant results, we aimed to ensure a comprehensive and accurate interpretation of the study findings while acknowledging the uncertainty associated with non-significant estimates.

Minor comments:

1. Page 4, line 8: The 86.7 million person-years seems wrong as the abstract (page 3, line 26) states that individuals were followed for 84.1 million person-years?

Answer: We thank the Reviewer for noticing this mistake in the “strengths and limitations” section. This has been corrected in Pages 3 and 11 to the correct value of 83.1 million person-years.

2. Please include in the description of the LISA which period this registry covers.

Answer: The period which LISA covers has been included in the methods sections. It now reads:

“Lastly, the Longitudinal Integration Database for Health Insurance and Labour Market Studies register (Swedish acronym, LISA), which had been compiled annually from 1990, and the Population and Housing Census (conducted every 5 years between 1960 and 1990), were used to obtain socioeconomic information.”

3. The definitions and data concerning the outcome are described before details concerning the exposure. The authors could consider describing the exposure before the outcome as the present study is a cohort study.

Answer: In line with the Reviewer’s suggestion, we have described the exposure before the outcome.

4. It is described under the exposure part how individuals were treated if they received both UC and Crohn’s Disease diagnoses. This section might benefit from a graphical illustration of different scenarios.

Answer: We thank the reviewer for this suggestion. We have taken this suggestion into consideration and have included a flowchart in the revised manuscript. The flowchart provides a visual representation of the screening process based on the inclusion and exclusion criteria, enhancing the clarity of the study methodology. Further, we attempted to clarify the classification of exposure in better detail.

“Furthermore, using data until the end of follow-up, individuals were excluded if they had an initial diagnosis of UC but subsequently received at least two CD diagnoses (n=10,682) or had a CD diagnosis not followed by at least two UC diagnoses (n=19,496; Figure 1 visually depicts the process of identifying the individuals included in the analysis).” (P6-7).

5. Page 9, line 14: Please consider changing the term “overall cohort” to “without UC” or similar.

Answer: We have made the suggested change. It now states:

“...for individuals without UC and for those with UC that did and did not undergo colectomy.”

6. Page 9, line 28: The cox-proportional hazards regression analyses were used for investigating the association between colectomy and development of dementia. What about the association between UC combined and risk of dementia?

Answer: We thank the reviewer for highlighting our unclear explanation. In our study, first, we investigated the association between UC and the risk of dementia, and then we further separated UC into with and without colectomy. We apologize for the lack of clarity in the sentence. This has been rewritten to reflect the analyses in the correct order. The manuscript now reads:

“First, we examined the association between UC (regardless of colectomy) and the risk of developing all-cause dementia. Subsequently, we separated the UC group into two subgroups based on colectomy status: those with and without a colectomy. By conducting separate analyses for each subgroup, we were able to assess associations with dementia risk for UC with and without colectomy.” (P8-9).